# Responses of *Melilotus officinalis* Growth to the Composition of Different Topsoil Substitute Materials in the Reclamation of Open-Pit Mining Grassland Area in Inner Mongolia

**DOI:** 10.3390/ma12233888

**Published:** 2019-11-25

**Authors:** Xinyu Kuang, Yingui Cao, Gubai Luo, Yuhan Huang

**Affiliations:** 1School of Land Science and Technology, China University of Geosciences (Beijing), 29 Xueyuan Road, Haidian District, Beijing 100083, China; 2112170021@cugb.edu.cn (X.K.); 2112170022@cugb.edu.cn (G.L.); 3012190017@cugb.edu.cn (Y.H.); 2Key Lab of Land Consolidation, Ministry of Natural Resources of the P. R. China, Beijing 100035, China

**Keywords:** land reclamation, soil reconstruction, topsoil substitute material, grassland mining area, Inner Mongolian grassland

## Abstract

The purpose of this study was to reveal that reconstructed soil composed of different types and proportions of materials has different effects on the growth of *Melilotus officinalis*, and to determine the most suitable formula of reconstructed soil materials to use for soil replacement. Using topsoil, coal gangue, fly ash, and rock and soil stripping materials from Shengli Mining Area of Inner Mongolia as raw materials, stratified and mixed pot experiments were carried out in a greenhouse using different proportions of each material. The differences in the aboveground biomass, leaf width, plant height, and root length of *Melilotus officinalis* plants in pot experiments were then compared using analysis of variance. The results showed that using different combinations of materials in different proportions affected the growth status of *Melilotus officinalis*, and their effects on biomass were greater than their effects on plant height, root length, and leaf width. When topsoil, coal gangue, and rock and soil stripping materials were mixed at a ratio of 3:3:4, respectively, the biomass of *Melilotus officinalis* increased by nearly 30% compared with that of plants potted in pure topsoil. When the content of coal gangue was controlled to be 30%, the content of fly ash was below 10%, and the content of rock and soil stripping materials was below 40%, the reconstructed soil conditions clearly promoted the growth of *Melilotus officinalis*. Coal gangue, rock and soil stripping materials, and fly ash can thus be used as substitutes for topsoil. Mixing soil reconstruction materials in the optimal proportion can solve the scarcity of topsoil in the grassland mining areas in the study region and, at the same time, can effectively improve the utilization of solid waste in this mining area.

## 1. Introduction

Coal resources have always been an important energy source in China, accounting for about 70% of the country’s primary energy production and consumption [1]. This country is the largest producer and consumer of coal in the world, and the coal output of China reached 3.52 billion tons in 2017, accounting for 46% of the world’s total output and 51% of the world’s coal consumption [2]. The regions of Shanxi, Shaanxi and Inner Mongolia account for 66.82% of China’s coal output. Open-pit coal mining has a greater efficiency, higher recovery rate, and lower cost than coal mining via underground mining. The percentage of mining in China that is done by open-pit mining has increased from 4% to 15% in recent years [3]. The open-pit coal mining areas in China are mainly distributed in desert, hilly, and grassland areas, including those in Shanxi, Shaanxi, and Inner Mongolia, which are ecologically sensitive [4]. While the large-scale exploitation of coal resources meets the needs of China’s economic construction, it also leads to a series of ecological and environmental problems [5], as well as social problems [6,7]. Compared with underground mining, the land damage caused by open-pit mining is more serious. In the process of open-pit coal mining, it is necessary to strip away all the of rock and soil layers above the coal seam, which will inevitably have large ecological and environmental impacts [8]. According to incomplete statistics, the area of land damaged by open-pit coal mining in China reaches 45,000 hm^2^ every year [9,10]. Therefore, it is imperative that the reclamation of damaged land in open-pit coal mining areas is vigorously promoted.

Soil reconstruction is the core component of land reclamation [11], and thus the quality of reconstructed soil directly determines the quality of land reclamation that is possible. The Quality Control Standard for Land Reclamation (TD/T 1036-2013) promulgated by the former Ministry of Land and Resources in 2013 stipulated that, in the northern grassland areas of China, the effective soil layer thickness should be greater than 30 cm when the purpose of reclamation is to recreate grasslands. In the process of open-pit mining, the platform-slope form of accumulation is often used in the construction dumping areas, which causes an increase in surface area that results in there being an insufficient amount of topsoil available to be used as soil cover. Soil is a key component in many ecosystem processes, such as nutrient circulation, water balance, and litter decomposition [12]. Soil structure and nutrient status are among the key indicators that are used to assess the restoration and maintenance of ecological functions in degraded ecosystems [13]. And Burley et al. [14,15,16] did several seminal works and important advances crossing top soil adaptation for vegetation growth. Therefore, the selection of topsoil substitutes has become a key part of the process of soil reconstruction in zones of topsoil scarcity.

The preferred substitutes for topsoil are generally industrial solid wastes from mining areas. Many laws and regulations concerning land reclamation in Western countries clearly stipulate that it is necessary to analyze the physicochemical properties and heavy metals content of the matrix of the soil and the overlying strata, and then select suitable substitutes for top soil before mining [17,18]. For example, in German reclamation, there are fewer topsoil sources, some of which come from the topsoil stripped from mining areas, and some from artificial soil [19]. In the United States of America (USA), when the amount of stripped topsoil is insufficient, it is necessary to find and screen suitable soil substitutes from the overburdened strata of coal seams. Baker et al. [20] obtained a kind of stable soil by mixing a fly ash series of combustion products and sludge compost into coal gangue (pH < 2), which could be directly applied in the field. Wilson-Kokes and Skousen [21] used weathered brown sandstone and un-weathered grey sandstone for soil reconstruction in an open-pit mine in West Virginia, USA. Through monitoring soil physical and chemical properties and measuring tree growth for eight consecutive years, that study found that the weathered brown sandstone soil had better physical and chemical properties, making it an ideal substitute material for topsoil. Inoue [22] introduced the application of fly ash as a topsoil substitute material in the land reclamation of an open pit mining area in Indonesia. Paradela et al. [23] tested and analyzed the basic characteristics of the slate powder produced in the open-pit mining process, and found that it could be used as a topsoil substitute material in the process of reclaiming open-pit mining. Domestic research on topsoil substitute materials in China started relatively late. Through pot and plot experiments, Ma et al. [24,25] found that the application of 20% fly ash in base soil was the best composition of substitutes for topsoil among those they tested to improve the yield and growth of soybean and maize. Working in an open pit mine in Inner Mongolia, Hu et al. [26] selected the sub-clay III layer of sub-clay as a topsoil substitute material through analyses of its physical and chemical properties seedling growth rate. On this basis, his team added peat [27], modified straw [28], vermiculite [29], and humic acid [30], and improved and optimized the formula of the substitute material to apply through assessing alfalfa growth and resisting performance. Because of the different ore-forming geological conditions that occur in different regions, the physical and chemical properties of mining solid wastes are different, so the proportion of solid wastes has strong regional characteristics. Thus, with the aim of solving regional problems, the best way to solve the problem of topsoil scarcity in mining areas is to select typical mining areas in which to carry out topsoil substitution material ratio tests, assess the impacts of the use of different reconstructed soil materials on the biomass of *Melilotus officinalis*, and determine the optimal ratio of different components to include in the topsoil substitute material [31]. 

Inner Mongolia is an important open-pit coal mining area in China, and it is also an ecologically sensitive area. The Shengli Mining Area in Inner Mongolia is located in the grassland region of eastern China, where the primary topsoil is relatively barren, which has become a serious bottleneck in the process of land reclamation in this area. This restriction causes a series of problems in land reclamation, such as insufficient soil cover thickness and poor vegetation growth. Therefore, the Shengli Mining area in Inner Mongolia was selected as a typical mining area for examination in the present study. This study was done with the aim of overcoming such obstacles in the process of land reclamation in the eastern grassland open-pit mining areas in Inner Mongolia; to do so, a method of reconstructing soil was put forward, starting from making full use of the common raw materials in the mining area; then combining the topsoil, coal gangue, fly ash, rock and soil stripping materials, and ash soil into different formulations of reconstructed soil; and then applying these in pot experiments. This was also done to reduce the cost of ecological land restoration and solve the environmental problems caused by the stacking of solid wastes, such as coal gangue and fly ash, as well as to provide a theoretical basis for soil reconstruction and the rational utilization of coal gangue and fly ash in the studied mining area.

## 2. Material and Methods

### 2.1. The Phase of Pot Experiment Design

In the pre-sampling process (Figure 1), we found that the thickness of the topsoil is about 20 cm in the undisturbed area, the following is the calcic horizon. The sampling of the reclaimed dumping site in the study area found that the thickness of the covering soil in most areas is about 10–15 cm. Owing to the randomness of construction, some areas will have a thickness of 5–10 cm. Therefore, there are two reconstruction modes, the layered mode and mixed mode. Four groups of schemes are set up in the layering mode. The purpose of determining the suitable plant growth with different soil cover thickness is to explore whether the plant can grow normally under this highly random construction condition. The comparison between topsoil and reconstructed soil is to explore whether the reclamation mode of reconstructed soil + coal gangue will have adverse effects on plant growth. The topsoil scheme is set as the control scheme, the topsoil scheme refers to the scheme in which only topsoil is used and no other substitute materials. The scheme is used as a control scheme. The topsoil used in the experiment is more than 20 cm soil collected in the field.

The following explains why *Melilotus officinalis* is suitable as a test species. First of all, this species is a legume herb with the effect of fertilizing the soil. Secondly, the characteristics of grass-tolerant, barren-resistant, and alkali-tolerant soils are highly adaptable to the region. Finally, in the reclamation work before, the grass raft was also planted and grown well in the study area. On the basis of the above three points, this species is suitable as a test species. Each group was repeated three times.

### 2.2. Proportion Determination of Mixing Method

Before conducting the experiments’ design (Figure 1), we consulted the relevant literature, combined with the test results of different materials, and had a preliminary understanding of different topsoil substitutes materials. Compared with topsoil, the rock and soil stripping is mainly attributable to poor nutrient status and large chunks of gravel; however, the stripping material is the material with the most similar physical properties to the surface soil. Therefore, in the reconstruction of soil, the content of stripping material should be controlled within a certain range. In the experiment, the content of stripping material is controlled below 60%. Coal gangue has a large particle size; it will cause a large loss of water, while the content of coal gangue is excessive, but coal gangue can improve the soil nutrient status. Therefore, the amount of gangue should not be excessive in the proportion. Fly ash is poor in nutrient status and is often used as a modifier. However, in order to fully demonstrate that the fly ash will cause poor plant growth, 10%, 30% and 60% are selected. It was also found that, under the condition that the content of fly ash was 60%, the leaves of the hibiscus were yellow and a large number of deaths occurred. This is also a prerequisite for our experimental design. Finally, on this basis, according to the texture, the control is in the range of loam after proportioning, according to different combinations and different proportioning tests. The experimental materials used in this study included topsoil (sandy loam soil), fly ash, coal gangue, and rock and soil stripping materials (a mixture of parent material and soil) (Table 1). The above experimental materials were obtained from the Shengli Mining Area of Inner Mongolia. A certain amount of ash soil was also used as fertilizer.

The experiment was carried out in the greenhouse of the China University of Geosciences (Beijing), China. The diameter and height of the flowerpots used were 20 cm and 22 cm, respectively, and the thickness of the reconstructed soil applied was 20 cm. On the basis of different reconstruction methods, the experimental scheme was divided into two groups, the layered variants schemes (Table 2) and the mixed schemes (Table 3), while an additional scheme, the total topsoil scheme (D1), was used as a control. Each scheme was repeated three times. A gradient test was set up to examine different mixtures of topsoil, rock and soil stripping materials, fly ash, and coal gangue, in different proportions, to form different reconstructed soils. Ash soil was used as a base fertilizer before the pot experiment, and was applied to a thickness of 2 cm per pot. To ensure uniform mixing, the required materials were poured onto a piece of canvas in turn according to the mixing ratio, and then the materials were turned from the bottom to the top by hand to mix them; this procedure was repeated at least five times until all of the materials were evenly mixed.

### 2.3. Monitoring of Indicators

Indicators are divided into two categories (Figure 1): biomass represents the overall growth status, and leaf width, leaf length, and root length represent the monomer growth status. The aboveground biomass of *Melilotus officinalis* grown in each pot was measured by harvesting. The aboveground parts of the potted vegetation were harvested and numbered in a ready-sealed bag. After that, the samples were dried until they reached a constant weight in an indoor oven at 65 °C, and then their weights were recorded.

The leaf width and total plant height of each *Melilotus officinalis* plant was measured with a ruler. Three *Melilotus officinalis* plants in each pot were selected for the measurement of these dimensions.

After harvesting the aboveground parts of *Melilotus officinalis* plants, the flower pot was cut longitudinally and the root system was collected. Six roots of different lengths were selected from each pot, and their lengths were then measured with a ruler. 

The experimental instruments used included a PL303 electronic balance (METTLER TOLEDO Instrument (Shanghai) Co., Ltd., Shanghai, China), a DHG-9245A electric heating blast dryer (Shanghai Yiheng Technology Co., Ltd., Shanghai, China), a ruler. The CP114 electronic balance is used to measure the biomass; the 101-2AB electric heating blast dryer is used to dry grass samples; the ruler is used to measure plant height, leaf width, and leaf length.

### 2.4. Data Analysis

All schemes are divided into two categories (Figure 1): the difference analysis in layered mode and the difference analysis in mixed mode. In the mixed mode, all schemes are divided into three groups, namely, fly ash group, coal gangue group, and rock soil stripping group. Take the fly ash group as an example, select the same or similar scheme of fly ash content, form a group together with the control scheme, and analyze the difference within the group. The difference analysis is divided into two parts: one is to analyze the general trend of the influence of some substitute materials on the growth difference of *Melilotus officinalis* under different content; the other is to analyze the difference of the proportion of other substitute materials on the growth of *Melilotus officinalis* under the same content of some substitute materials. The data for the growth status of *Melilotus officinalis* were analyzed by one-way analysis of variance (ANOVA) in SPSS 20.0 (IBM SPSS Statistics, Chicago, IL, USA). Through the results of the difference analysis, the range value of the content of each topsoil substitute material in the reconstructed soil was determined, and the best formula of the reconstructed soil was obtained in the ratio test.

## 3. Results

### 3.1. Differences in Plant Growth Indicators among the Layered Variants Schemes

There were no significant differences in the leaf width indicator between the layered variants schemes and the control scheme, and there were no significant differences in this indicator among any of the layered variants schemes (Figure 2). The layered accumulation method of soil reconstruction (i.e., using a 1:1 mixture of topsoil and geotechnical stripped matter) from topsoil and coal gangue did not have a significant impact on leaf width. For the plant height indicator, only C1 (5 cm of topsoil + 15 cm of coal gangue) significantly differed from the control scheme, and there were no significant differences between the other layered variants schemes and the control scheme, nor among the four layered variants schemes. For the root length indicator, C2 (5cm of reconstructed soil + 15cm of coal gangue), C4 (10cm of reconstructed soil + 10cm of coal gangue), and the control scheme were all significantly different from each other, while C1, C3 (10 cm of topsoil + 10 cm of coal gangue), and the control scheme did not significantly differ, and there were no significant differences among the four layered variants schemes. In terms of the biomass indicator, only C1 and C4 significantly differed, and there were no significant differences among the other treatments. The biomass of *Melilotus officinalis* ranged from 2.20 g/pot to 3.54 g/pot, and that of *Melilotus officinalis* grown in C4 (3.54 g/pot) was higher than that of those plants grown in pure topsoil (3.48 g/pot). In the layered variants schemes, only the biomasses of plants grown in C4 and C1 were significantly different. The biomass of plants grown in a mixture of topsoil with geotechnical stripping material in the upper layer was greater than that of those grown with only pure topsoil in the upper layer. 

This shows that the growth needs of *Melilotus officinalis* be met when the thickness of the overlying soil is above 5 cm, but *Melilotus officinalis* grows even better when the thickness of the overlying soil is 10 cm.

### 3.2. Differences in Plant Growth Indicators among the Mixed Schemes

#### 3.2.1. Differences in Plant Growth Indicators among Mixtures with Different Fly Ash Contents

The leaf width indicator of plants grown in schemes in which 10% of the topsoil was replaced with 10% fly ash was either 10.56 mm or 8.78 mm (Figure 3). The results of ANOVA showed that there were significant differences between these two schemes. At the same time, there was no significant difference in this indicator between H3 and D1, but there was a significant difference between H16 and D1. The main difference between the material used in H3 and H16 was that in H3, the same amount of topsoil was replaced with 10% coal gangue, which shows that a certain amount of coal gangue can promote the growth of leaves. Under the treatment with a 60% fly ash content, the width of the leaves in H15 was 8.33 mm, which was significantly different from that of the leaves in D1, H3, and H16. The root length of plants grown in the schemes with 10% fly ash content was either 13.46 cm or 14.69 cm. The results of ANOVA showed that there was no significant difference in root length between these two schemes, and there was no significant difference in root length between these two schemes and the control scheme. Under the condition with a 60% fly ash content, the length of roots in H15 was 10.94 cm, which was significantly different from that of roots in the control scheme and H3, but not from that in H16.For the plant height indicator, the plant height of *Melilotus officinalis* in H3 was 16.22 cm when grown in material with a 10% fly ash content, which was significantly higher than that of plants grown in the D1 scheme (11.56 cm) and H16 (10.67 cm), although there was no significant difference between the plant height in H16 and that of the control plants. Further, the height of plants in H15 was 11.11 cm with a 60% fly ash content, and there was no significant difference between the plant height in H15 and the control plant height, nor was there a significant difference in plant height between H15 and H16. However, the plant height in these conditions was significantly lower than that in H3. Under the schemes with 10% fly ash content, the plant biomass was either 2.71 g/pot or 1.95 g/pot. The results of ANOVA showed that there was no significant difference between the biomass of plants grown in H3 and that of those grown in D1, H16, and D1, but there was a significant difference between the biomass of plants grown in H3 and D1. Under the scheme with a 60% fly ash content in the soil, the biomass was 1.57 g/pot, which was significantly lower than that in the control scheme and H3, but not significantly different from the biomass in H16. In conclusion, there were no significant differences in leaf width, root growth, and biomass between H3 and D1, but plant height was significantly higher in these conditions than in D1. H16 was not significantly different from the control in terms of plant height and root length, but did result in significantly smaller leaf width and biomass values, and H16 was significantly smaller than H3 in terms of leaf width and plant height. H15 was significantly smaller than the control in terms of all indicators except plant height; significantly different in terms of leaf width, plant height, and biomass from H3; and only significantly different in terms of root length from H16.

When the content of fly ash in the topsoil material was 30%, the width of *Melilotus officinalis* leaves was 10.00 mm, 10.56 mm, or 9.11 mm; the root length was 15.39 cm, 16.39 cm, or 13.67 cm; the plant height was 14.44 cm, 15.11 cm, or 12.67 cm; and the biomass was 2.45 g/pot, 2.45 g/pot or 1.95 g/pot (Figure 4). In terms of the leaf width indicator, only that in H17 and H5 was significantly different, but there were no significant differences in this indicator among any of the other schemes. Comparing H17 with H5, the materials used in these schemes differed in that 30% of the topsoil in H17 was replaced with the same amount of rock and soil stripping material. In terms of root length, there were no significant differences between any of these three schemes and the control scheme, and there were no significant differences among the three schemes. However, the plant height in all three of these three schemes was greater than that in the control scheme. The results of ANOVA showed that the plant height was significantly higher in H4 and H5 than in the control scheme. There was no significant difference between the plant height in H17 and that inD1, nor was there any significant difference in this indicator between H4 and H17, or between H5 and H17. The results of the ANOVA of biomass data showed that the biomass of plants grown in all three schemes was significantly lower than that of plants grown in the control scheme, but there were no significant differences in biomass among these three schemes. 

The results of the ANOVA done for all four indicators (Figure 4) showed that H4 and H5 did not perform equally to D1 in terms of the leaf width, root length, and plant height indices, while H17 did not perform worse or better than the control (D1) in terms of root length and plant height, although the biomass in this treatment was significantly lower than that in the control when the fly ash content of the topsoil material was 30%. 

Generally speaking, H3 was the best among the schemes that were tested with fly ash at promoting the growth of *Melilotus officinalis*, and the other schemes were obviously worse than the control scheme in terms of the biomass indicator. When the content of fly ash in the soil material was 30%, the performance of each scheme was not worse than that of the control scheme, except for the biomass indicator. However, when the content of fly ash was 10%, the biomass performance was slightly larger. These results showed that if fly ash is used as a substitute material for topsoil, it should not comprise too high of a percentage of the formula, and its effect is better when it is used along with coal gangue.

#### 3.2.2. Differences in Plant Growth Indicators among Mixtures with Different Contents of Rock and Soil Stripping Materials

Under the conditions in which the content of rock and soil stripping materials in the topsoil material was 20%, the width of *Melilotus officinalis* leaves was 9.56 mm, 10.56 mm, or 8.00 mm the root length was 14.48 cm, 10.72, cm or 15.18 cm; the plant height was 11.44 cm, 12.33 cm, or 8.39 cm; and the biomass was 2.67 g/pot, 2.06 g/pot or 3.41 g/pot (Figure 5). The results of ANOVA showed that there was a significant difference between the leaf width in H12 and those in H8, H10, and D1, but no significant differences among those in H8, H10, and D1. Plant height in H12 and H8 significantly differed, as did that in H10 and D1, but there were no significant differences in plant height among H8, H10, and D1. Plant height in H10 and H8 significantly differed, as did that in H12 and D1, while there were no significant differences among H8, H12, and D1. There were significant differences in biomass among H10, H12 and D1, but no significant differences among H8, H12 and D1. At the same time, there were no significant differences among H8, H12 and D1. Generally speaking, the performance of H8 in terms of all four indicators was not different from that of the control scheme. The H10 scheme resulted in poor root growth and biomass accumulation, and H12 resulted in narrower leaves and lower plant heights.

The results of ANOVA showed that there were no significant differences among H5, H7, H9 and D1 when the content of rock and soil stripping materials in the soil mixture was 30%, and there were no significant differences among these three schemes (Figure 6). In terms of plant height, that in H5 was significantly higher than that in the other three schemes, but there was no significant difference among these three schemes. The results showed that when the content of rock and soil stripping materials included was maintained at 30%, all four growth indicators of *Melilotus officinalis* growth examined showed that the growth of this plant was good in these conditions.

When the content of rock and soil stripping materials was 40%, the plant height in H13 was significantly lower than that in the control scheme, while the root length in H14 was significantly less than that in the control scheme (Figure 7). In terms of the other indicators, there were no significant differences between the two schemes and the control scheme, and there were no significant differences in the four indicators in H11 from their values in the control scheme. The results showed that in H13, plant height was significantly lower than that in H14; in H14, root length was significantly less than that in H11 and H13; and in H11, the plant biomass was significantly higher than that in H13 and H14.

When the content of rock and soil stripping materials was 50%, only the plant height in H6 was significantly lower than that in the control scheme (Figure 7). Compared with the schemes containing 40% rock and soil stripping materials, the leaf width and root length of plants in H6 were significantly worse than those in H13, but not significantly different from those in H11 and H14. The height of plants grown in H6 was significantly lower than that in H14, but not significantly different from those in the other two schemes. Meanwhile, the biomass of plants in H6 was significantly less than that of plants grown in H11.

Generally speaking, when the content of rock and soil stripping materials in the topsoil mixture was 40%, H11 performed the best, while the other two schemes performed poorly in terms of some indicators. The difference in the formulations of these three schemes lies in their relative content of coal gangue and surface soil. When the content of rock and soil stripping materials was 50%, the performance of H6 was still acceptable. Although the only significant difference from the control scheme found was in terms of plant height, the performance of the other indicators examined was generally fair.

When the content of rock and soil stripping materials in the topsoil mixture was 60%, the ANOVA results showed that there were significant differences in all growth indices among all of the schemes, except for the root length indicator (Figure 8). The width of leaves in H3 was significantly greater than that in the other schemes, but not significantly different from that in the control scheme. The leaves of plants grown in H1 and H2 were significantly narrower than those of plants grown in D1, but not significantly different from those in any of the other schemes. The widths of the leaves in H16, H17, and H18 were not significantly different from that in the control scheme. The plant height in H3 was significantly greater than that in the other groups, but also greater than that in the control group. Only the plant height in H2 was significantly lower than that in the control group. There were no significant differences in plant height between H1, H16, H17, and H18 and D1. The biomass of plans grown in H1, H2, and H3 was not significantly different from that of those grown in D1. The biomass in the other three schemes was significantly lower than that in the control scheme, and also significantly lower than that in the H1 scheme. There were no significant differences between the biomass in H2 and H3 and those in the other schemes. When the content of rock and soil stripping materials was 60%, H3 performed better in terms of all four indicators, and achieved plant heights that were even significantly higher than those measured in the control scheme. H1 performed poorly in terms of the leaf width indicator, but performed better in terms of the other three indices measured. 

Generally speaking, the content of rock and soil stripping materials in the topsoil substitute mixture had little effect on the leaf width, plant height, and root length of *Melilotus officinalis*, but had some impact on its biomass. This was especially true when the content of rock and soil stripping materials was between 40% and 60%, which resulted in the biomass of more than half of the schemes tested being significantly less than that in the control schemes. At the same time, it was found that the scheme in which the highest biomass was observed, H11, contained 40% rock and soil stripping materials, which indicated that the content of rock and soil stripping materials in the formula should be maintained below 40%.

#### 3.2.3. Differences in Plant Growth Indicators among Mixtures with Different Coal Gangue Contents

When the coal gangue content of the topsoil mixture was 15%, ANOVA results showed that the leaf width, plant height, and biomass in H2 were significantly worse than those in D1, while the root length was not significantly different between these schemes (Figure 9). Among the schemes with a gangue content of 10%, it was found that the width of the leaves in H2 was obviously less than that in H3, H4, and H5, but not significantly different from that in H17 and H18. The height of plants in H2 was significantly less than that in the other schemes with a 10% gangue content. The length of the roots in H2 was only significantly shorter than that in H5, which was basically the same as that in the other schemes. The biomass of plants grown in all of the schemes with a 10% gangue content was almost the same, and these differences were not significant. When the gangue content of the topsoil mixture was 10%, the ANOVA results showed that there were no differences between H3, H4, H5, H17, and H18 and D1 in terms of leaf width and root length. Meanwhile, those in H3, H4, H17 and H18 were significantly higher than those in D1, and those in H17 and H18 were basically the same as that in D1. Only the biomass in H3 was not significantly different from that in the control scheme, and the biomass in the other schemes was smaller than that in the control scheme. The results showed that the leaf width and plant height in H3 and H4 were significantly larger than those in H17 and H18. There was no significant difference between the leaf widths in H4 and H18, but the plant height in these schemes was significantly higher than that in H18. Meanwhile, there was no significant difference in root growth and biomass between these schemes.

Overall, only the biomass in H3 did not differ significantly from that in D1. In terms of leaf width and plant height, H3 showed better performance than the other schemes, and the plant height in this case was even significantly greater than that observed in D1. The performance of H2 in terms of leaf width, plant height, and biomass was smaller than that of the control. The performances of H4, H5, H17, and H18 in terms of leaf width, plant height, and root length were not worse than those of the control (D1), but the biomass was lower in these schemes than that in the control.

When the coal gangue content was 20%; there were no significant differences in the root growth and biomass in H6, H7, and H14 compared with those in the control scheme; and there were no significant differences among these three schemes (Figure 10). There were no significant differences in leaf width between these three schemes and the control scheme (D1), but the leaf width in H7 was significantly greater than that in H6 and H14. The plant height in H7 was not significantly different from that in the control. The plant height in H6 and H14 was significantly lower than that in H7 and D1, and the plant height in H6 was significantly lower than that in H14. Among the three schemes, the performance of H7 in terms of all four growth indicators was not inferior to that of D1, H6 or H14. When comparing H7 with the other two schemes, it was found that the topsoil mixtures in the other two schemes both had a higher content of geotechnical debris than that in H7, but contained less topsoil.

When the gangue content was 30%, the variance analysis showed there were no significant differences among H8, H9, H11 and D1, and there were no significant differences among these three schemes (Figure 11). The results showed that when the content of coal gangue was maintained at 30%, all four growth indicators of *Melilotus officinalis* growth examined showed that the growth of this plant was good in these conditions.

When the gangue content was 40%, the variance analysis showed that the biomass of H10 and H13 was significantly lower than that of control D1, and the difference of leaf width and D1 was not significant, while H10 and H13 were not significant in these two indicators; H13 plant height was significantly lower than that of D1 and H10; H10 plant height and D1 plant height had no significant difference; and H10 root length was significantly shorter than that of H13 and D1.

When the gangue content was 40%, the ANOVA results showed that the biomass in H10 and H13 was significantly lower than that in the control (D1), and the leaf width in these schemes was not significantly different from that in D1, while H10 and H13 did not significantly differ in either of these two indicators. Plant height in H13 was significantly lower than that in D1 and H10, although the plant height in H10 and that in D1 were not significantly different. The root length in H10 was significantly less than that in H13 and D1. 

When the gangue content was 50%, the results of ANOVA showed that there was no significant difference between plant biomass and root length in H12 and those in D1, and the leaf width and plant height in this scheme were significantly larger than those in D1 (Figure 12). The biomass in H12 was greater than that in H10, but not significantly different from that in H13, while the leaf width was less in H12 than in H10 and H13. Plant height in H12 was significantly less than that in H10, but not significantly different from that in H13. The root length in H12 was significantly greater than that in H10, but the difference of H12 from H13 in terms of this indicator was not significant.

In general, when the content of coal gangue in the topsoil mixture was 20%–30% or 10% or less, the growth condition of *Melilotus officinalis* was better than that with higher coal gangue contents. This was especially true when the coal gangue content was controlled at 30%, at which there were no significant differences between any of the growth indicators of *Melilotus officinalis* and those in the control scheme. When the gangue content was 20%, the plant biomass was not significantly different from that in the control scheme, but the leaf width and plant height were significantly smaller than those in the control scheme.

## 4. Discussion

### 4.1. Reasons for the Differences in the Physicochemical Properties of Reconstructed Soils and the Growth Status of Melilotus officinalis

The theory of soil-forming factors posits that soil is the product of multiple natural factors, such as biology, climate, parent material, topography, and time, as well as human activities. In this study, on the basis of the theory of soil-forming factors and the reconstruction of soil parent material, different soil profiles were reconstructed using solid wastes (coal gangue, rock and soil stripping materials, and fly ash) generated during mining as surface soil substitutes. Dong et al. [32] have shown that the physicochemical properties of soils that developed from different parent materials are quite different. For instance, coal gangue has a large particle size and high organic matter content [33,34]; fly ash has a large particle size [35], is hydrophilic [36], and has poor nutrient status [37]; and the physical properties of rock and soil exfoliates from stripping are similar to those of the original topsoil, but with a lower organic matter content, as the organic matter content, total nitrogen content, available phosphorus, and available potassium content of the rock and soil stripping are more than double that of the topsoil. Therefore, the physicochemical properties of reconstructed soils vary greatly if different combinations of materials are used, for example, the organic matter content of different schemes vary greatly (Table 4). 

The effects of including materials in different ratios on the production of plant biomass were greater than those of these differences on leaf width, plant height, and root length. During the experiment, it was found that the mixture schemes resulting in lower biomass tended to have fewer plants and lower coverage, but the growth of a single plant in these cases was sometimes better; further, the schemes resulting in lower biomass but more plants tended to have poor performance in terms of leaf width or plant height. The reason for this result is that there were differences in the proportions of different components included in the soil substitute in these different schemes, resulting in differences in their physicochemical properties. Different endowments of soil resources result in different productivity of the replacement soil. Therefore, in the actual process of vegetation reconstruction, the use of a reasonable planting density and topdressing will also play important roles in the successful growth of *Melilotus officinalis*.

The number of plants in the potted plants is not indicated in the text. This result is derived from the data collected by the camera. The number of plants can be seen from the photographs, so this conclusion is reached. However, this result is more because of our subjective knowledge.

### 4.2. Reasons for the Better Growth of Melilotus officinalis under Specific Substitute Material Ratios

When the coal gangue content of the mixture used was below 30% or 10%, the growth of *Melilotus officinalis* performed better in terms of all four indicators assessed. According to the physical and chemical properties of solid waste, the average organic matter content of coal gangue is 43.90 g/kg, which is about 25% higher than the organic matter content of topsoil, and the available K content of coal gangue is about 10% higher than that of topsoil. Coal gangue has a large particle size and large pores; its particle size is 2–5 cm. When stacked, there are many and large pores between coal gangue particles. This characteristic means that, when coal gangue is piled up, moisture can easily seep downward through it [38], so the content of coal gangue in the soil replacement material should not be too high. However, including coal gangue can improve the soil nutrient status [39]. At the same time, as the degree of weathering increases, the sizes of the internal weathering cracks in coal gangue increase, which makes coal gangue have more water-holding capacity [40], and thus be more conducive to the growth of vegetation. However, in the actual production process, coal gangue is generally excessively produced, whereas the availability of the surface soil is limited, so the optimal amount of coal gangue used should be about 30%. When layered and overlapped coal gangue and reconstructed soil or topsoil is used, the roots of *Melilotus officinalis* can penetrate into the coal gangue layer; the root length of *Melilotus officinalis* is about 11–12 cm. The development of crevices in the coal gangue layer can provide the necessary water and nutrients for *Melilotus officinalis*, helping this plant to grow better.

Fly ash is mainly used to improve the physical structure of clay because of its large particle size [41], but clay did not exist in the sandy loam soil used in this study’s experiments. The hydrophilicity of fly ash has little effect on plant growth under conditions in which an adequate supply of the water required for the growth of vegetation is available or provided. Fly ash can reduce soil bulk density, increase porosity, adjust the three-phase ratio, and increase ground temperature. If used to improve sandy soil, it can increase water holding capacity and hydraulic conductivity, which is helpful to prevent crust. However, owing to the fact that there is almost no nitrogen in fly ash, it was also found that the leaves of *Melilotus officinalis* began yellowing during the experiment when this material was included. According to Liebig’s law of the minimum, this characteristic thus became a limiting factor for the growth of *Melilotus officinalis*. That is to say, under the conditions of an indoor pot experiment, as the fly ash content of the mixture increases, the soil nutrient condition becomes worse, resulting in lower production of *Melilotus officinalis* biomass. In 2008, Chen et al. [42] also showed that, with an increase in fly ash consumption, the growth of plants deteriorated.

The use of weathered fly ash can increase the accumulation of Se in crops [43]. Some short-term indoor incubation experiments have found that adding un-weathered fly ash to sandy soil can inhibit microbial respiration, enzyme activity, and soil N cycling [44,45]. Fly ash contains 5–30% toxic elements; especially Cd, Cu and Pb can be filtered out, which may cause soil, water, and biological pollution, especially the high content of soluble salt in weathered fly ash, which is more likely to cause groundwater pollution [46]. 

However, according to the pot experiment results of this study, when the content of fly ash was below 10%, it had little effect on the growth of *Melilotus officinalis*.

### 4.3. Benefits and Limitations of Using Solid Waste from the Mining Industry as a Soil Reconstruction Material

The selection of a reasonable compositional scheme based on experiments using the different solid waste produced in mining processes as soil substitute materials is of great significance for land reclamation in mining areas with scarce topsoil. Firstly, this allows the problem of poor vegetation growth caused by the insufficient thickness of the overlying soil in such areas to be solved. Secondly, the cost to mining enterprises of purchasing topsoil is greatly reduced, as reconstructing the soil using substitute materials can save more than 50% of the available topsoil; for example, assuming a soil cover thickness of 30 cm, the total depth of the surface soil needed is thus less than 1500 m^3^ per hectare, and if the local price of surface soil is about 30 yuan/m^3^, then in this case, the reclamation investment cost is reduced by 45,000 yuan per hectare. Finally, the problem of the disposal of solid waste generated in the mining process is solved. It can thus be seen that using solid waste as a substitute material for topsoil has certain economic and ecological benefits. However, at the same time, the current study’s results are based on laboratory tests, so their applicability in the field is not yet known. The next step is to carry out plot experiments in the field to determine the best application scheme. The pot experiment can only simulate the conditions of field crop growth to the maximum extent, it cannot achieve complete consistency, so it will cause differences in crop growth. In the pot experiment, the mixing of different materials is more uniform, but in field work, this effect can not be achieved. In the field experiment, we found that there may be only one material in some areas, but no mixing. In the actual construction process, the compaction of reconstructed soil by large-scale machinery will result in the increase of bulk density and the decrease of porosity, which will inevitably affect the growth of plants.

## 5. Conclusions

(1) On the basis of the present study’s results, the use of mining solid wastes as a substitute for topsoil to sustain plant growth appears to be feasible according to analyses of the four selected indices of the growth status of *Melilotus officinalis* tested. This conclusion can help the owner of the mining area solve the problem of resource utilization of solid waste and scarcity of topsoil in similar areas.

(2) When the thickness of the reconstructed upper soil was greater than 10 cm, the biomass of *Melilotus officinalis* was higher than that obtained with pure topsoil, and when the reconstructed upper soil was placed above the natural soil, the biomass of *Melilotus officinalis* obtained was higher than that obtained on topsoil alone. When the amount of coal gangue added was controlled to be 30% of the mixture, the biomass of *Melilotus officinalis* obtained was the best, and the values of other growth indicators were also better. The overall biomass obtained with the mixture schemes containing fly ash was lower, and was obviously different from that obtained in the control scheme, but the growth of individual plants was better; therefore, the growth of *Melilotus officinalis* was better when the amount of fly ash added was controlled to be 10% or less. When the content of rock and soil stripping materials from mining was controlled to be 40% or less, *Melilotus officinalis* showed good growth in terms of all four indicators. This scheme was thus more suitable for sustaining the growth of *Melilotus officinalis* than all of the others tested. Also, when the ratio of topsoil, coal gangue, rock and soil stripping materials was 3:3:4, respectively, the biomass of *Melilotus officinalis* was the highest, and the increases in leaf width, plant height, and root length were also better. However, this conclusion comes from laboratory tests, and it is unknown whether it is applicable to the local area. This result only shows that these materials can be used as substitute materials for topsoil. In different mining areas, the specific proportion needs to be tested before it can be obtained.

(3) The biomass of *Melilotus officinalis* significantly differed among the schemes tested, but the other three indicators did not significantly differ among schemes in many cases. In treatments in which plants had small biomass and also tended to produce fewer plants, the growth statuses of individual plants were better. Treatments that produced average amounts of plant biomass, but more plants tended to have poor performance in terms of leaf width or plant height. Therefore, in the actual reclamation process, planting density should be reasonably arranged according to the physical and chemical properties of the soil. Of course, the best way is to reclaim land strictly according to the best proportion obtained from the experiment, but the economic and ecological conditions of different mining areas are different, so it is likely that the best proportion cannot be chosen. Therefore, the combination of substitute materials proportion and planting method can achieve the goal of reclamation better.

## Figures and Tables

**Figure 1 materials-12-03888-f001:**
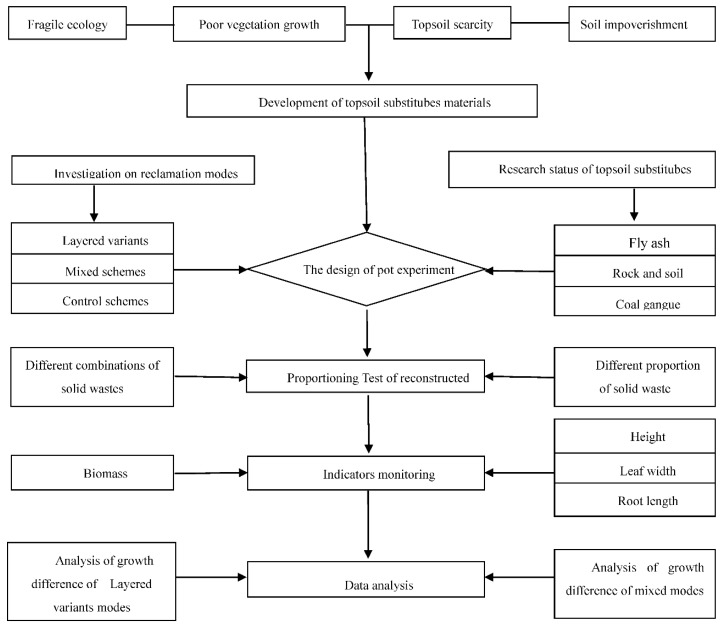
Methodological diagram.

**Figure 2 materials-12-03888-f002:**
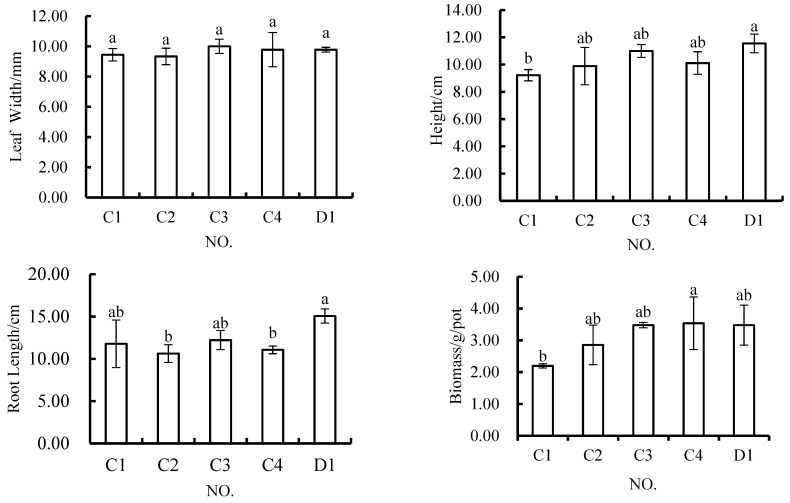
In *Melilotus officinalis*, growth indicators with different soil substitute materials applied under the layered variants schemes. Note: Different letters denote significant differences at the 0.05 level.

**Figure 3 materials-12-03888-f003:**
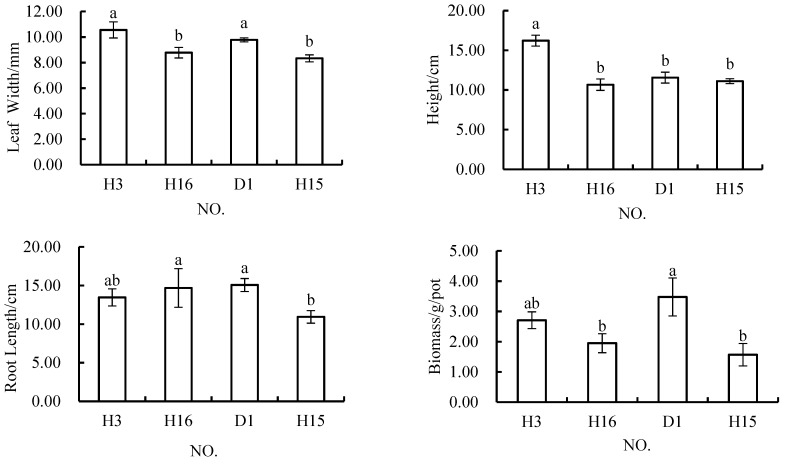
Differences in *Melilotus officinalis* growth indicators among soil replacement mixtures containing 10% and 60%.

**Figure 4 materials-12-03888-f004:**
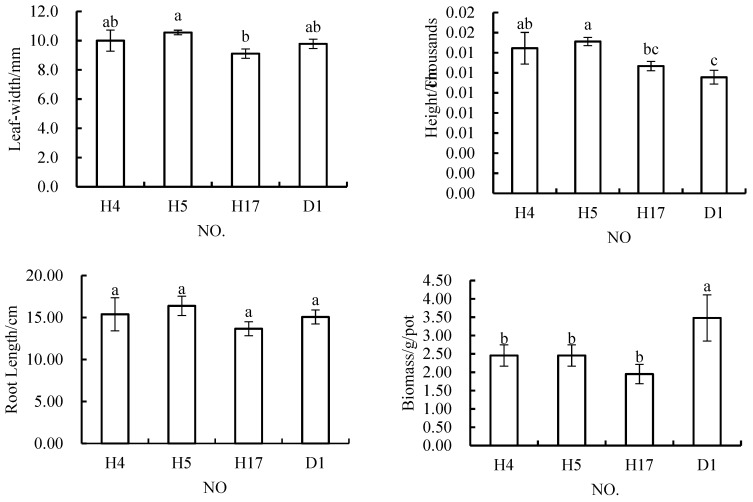
In *Melilotus officinalis*, growth indicators among soil replacement mixtures containing 30% fly ash.

**Figure 5 materials-12-03888-f005:**
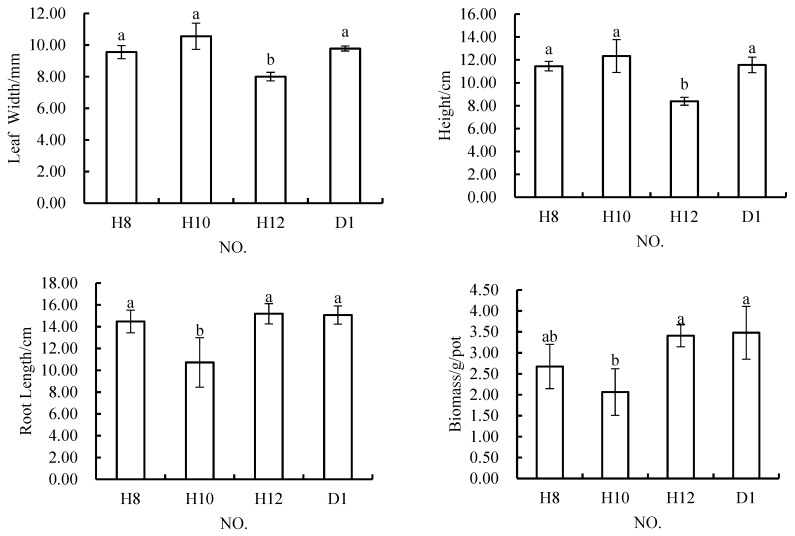
In *Melilotus officinalis*, growth indicators among soil replacement mixtures containing 20% rock and soil stripping materials.

**Figure 6 materials-12-03888-f006:**
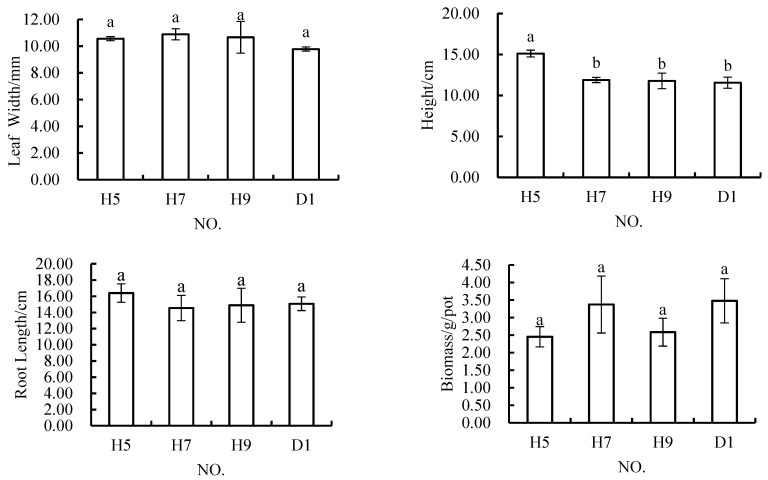
In *Melilotus officinalis*, growth indicators among soil replacement mixtures containing 30% rock and soil stripping materials.

**Figure 7 materials-12-03888-f007:**
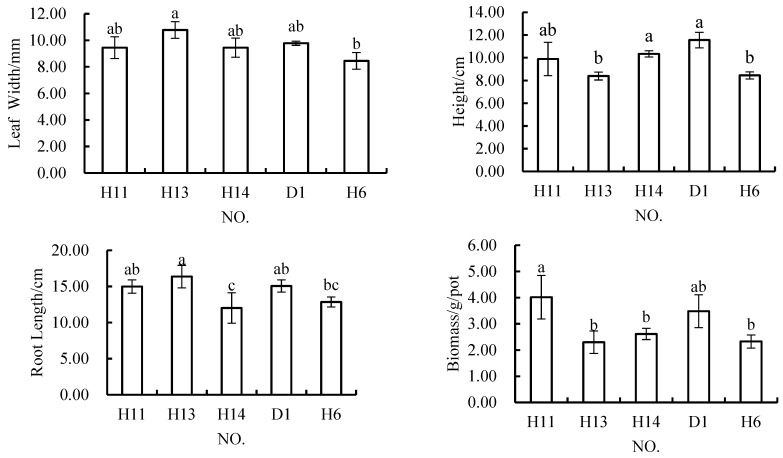
In *Melilotus officinalis,* growth indicators among soil replacement mixtures containing 40% and 50% rock and soil stripping materials.

**Figure 8 materials-12-03888-f008:**
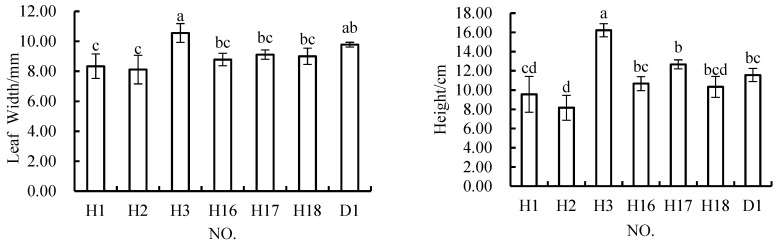
In *Melilotus officinalis*, growth indicators among soil replacement mixtures containing 60% and 50% rock and soil stripping materials.

**Figure 9 materials-12-03888-f009:**
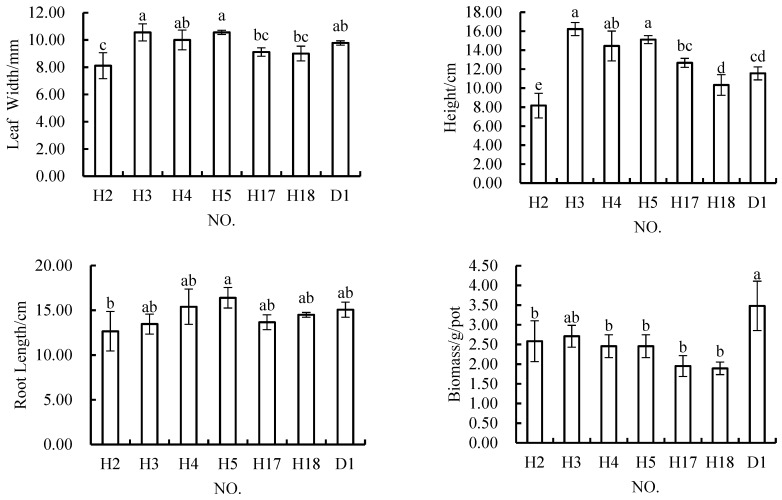
In *Melilotus officinalis,* growth indicators among soil replacement mixtures containing 10% and 15% coal gangue.

**Figure 10 materials-12-03888-f010:**
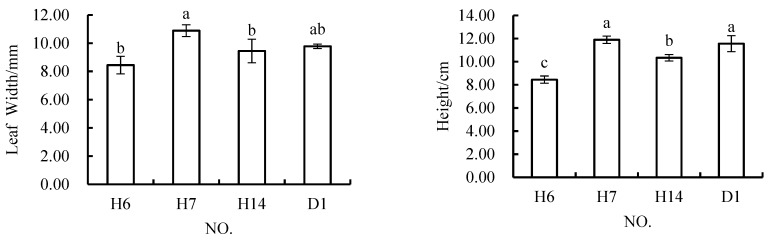
In *Melilotus officinalis*, growth indicators among soil replacement mixtures containing 20% coal.

**Figure 11 materials-12-03888-f011:**
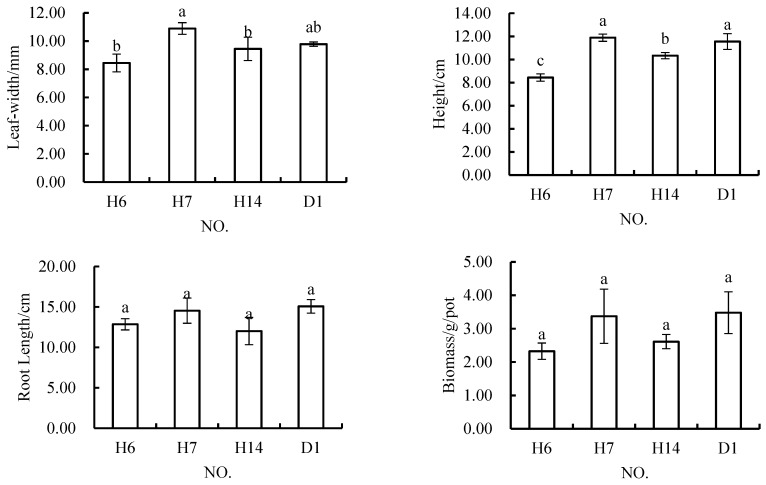
In *Melilotus officinalis,* growth indicators among soil replacement mixtures containing 30% coal gangue.

**Figure 12 materials-12-03888-f012:**
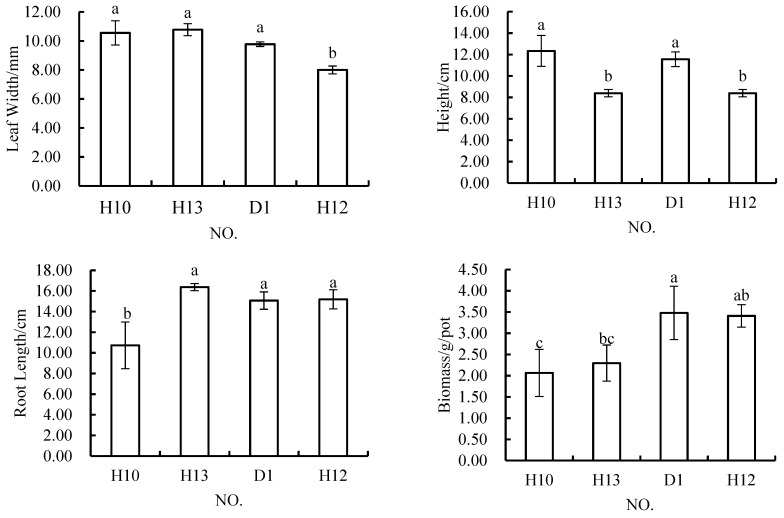
In *Melilotus officinalis,* growth indicators among soil replacement mixtures containing 40% and 50% coal gangue.

**Table 1 materials-12-03888-t001:** Background physicochemical properties of the different materials tested.

Materials	Organic Matter (g·kg^−1^)	Total N (g·kg^−1^)	Available P (mg·kg^−1^)	Available K (mg·kg^−1^)	Texture/Particle Size
Topsoil	35.4 ± 7.4	2.0 ± 0.5	5.30 ± 2.17	133.00 ± 54.30	Sandy loamy soil
Coal gangue	43.9 ± 34.0	0.7 ± 0.4	2.63 ± 1.01	145.83 ± 92.83	2–5cm
Fly ash	________	_______	_______	________	_______
Rock and soil stripping material	18.8 ± 5.1	0.9 ± 0.3	1.83 ± 0.86	50.97 ± 15.43	Sandy clay loam

Note: Fly ash is a byproduct of coal combustion. After combustion, most of the retained chemical elements, such as Si and Al, are the main nutrient elements, as well as trace elements, including As and Mn.

**Table 2 materials-12-03888-t002:** List of the layered variants potting experimental schemes applied.

NO.	Surface Material	Thickness (cm)	Bonding Material	Thickness (cm)
C1	Topsoil	5	Coal gangue	15
C2	Topsoil + Wastes	5	Coal gangue	15
C3	Topsoil	10	Coal gangue	10
C4	Topsoil + Wastes	10	Coal gangue	10

**Table 3 materials-12-03888-t003:** List of the mixed potting experimental schemes applied.

NO.	Topsoil (%)	Coal Gangue (%)	Rock and Soil Stripping (%)	Fly Ash (%)
H1	40	0	60	0
H2	25	15	60	0
H3	20	10	60	10
H4	60	10	0	30
H5	30	10	30	30
H6	30	20	50	0
H7	50	20	30	0
H8	50	30	20	0
H9	40	30	30	0
H10	40	40	20	0
H11	30	30	40	0
H12	30	50	20	0
H13	20	40	40	0
H14	40	20	40	0
H15	40	0	0	60
H16	30	0	60	10
H17	0	10	60	30
H18	30	10	60	0

**Table 4 materials-12-03888-t004:** The organic matter content of different schemes.

**NO.**	**H1**	**H2**	**H3**	**H4**	**H5**	**H6**
Organic Matter (g·kg^−1^)	3.87	6.91	10.73	11.37	9.07	7.46
**NO.**	**H7**	**H8**	**H9**	**H10**	**H11**	**H12**
Organic Matter (g·kg^−1^)	8.12	11.34	10.43	17.92	7.15	15.16
**NO.**	**H13**	**H14**	**H15**	**H16**	**H17**	**H18**
Organic Matter (g·kg^−1^)	13.05	8.69	3.28	3.91	8.80	3.64

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
