# Peer review of "Responses of Melilotus officinalis Growth to the Composition of Different Topsoil Substitute Materials in the Reclamation of Open-Pit Mining Grassland Area in Inner Mongolia"

_materials, 2019, doi:10.3390/ma12233888_

Round 1

Reviewer 1 Report

Generally, this study presents relevant results of an investigation on reclamation measures. The question of fertile topsoil layers for post-mining landscapes is an issue in mining regions worldwide and different solutions have been developed to overcome existing problems. However, the manuscript shows a number of serious shortcomings. First of all the language of the manuscripts needs a rigorous editing. Several terms need to be checked and many words need to be separated. Further, the described experiments need to be explained further in detail. E.g., the selection of the test plant species remains unclear: Is this species a native species of the described mining region in China and has it specific functions for reclamation? Can this species be regarded as exemplary for other species of the region?

In detail, additional questions arise:

l. 47: “well” mining: do you mean subsurface mining? l. 75: as I know the conditions in the Rhenish Lignite Mining District very well, I found that the reference is wrongly cited: Loess deposits in Germany are of Quaternary age and must be considered as natural topsoil material, not as substitute of topsoil. l. 95: “iste”? This term is used repeatedly throughout the text. l. 99: (see above), please explain here properly why this species is suitable as test species Tab. 1: please provide information about pH values of the substrates and of heavy metal contents. Particularly fly ash is usually problematic due to high pH values and higher heavy metal contents. In addition, please explain why fly ash had a sandy texture in your experiments. Usually fly ash is characterized by high amounts of silt (particles which are able to fly very well) l. 127: please explain for which purposes these devices were used in your experiments or for laboratory analyses. l. 129 ff: Please discuss if the applied very shallow layers can be seen as representative or as comparable with field conditions. It can be assumed that roots are growing through such thin layers so that the layering does not make much sense. I would also expect that it will not be possible to create such thin layers under field conditions. l. 133: why do you call the layered variants “hierarchical”? I think this term does not fit here. l. 134: the “topsoil scheme” is not further explained. Please provide some basic information on the topsoil used for this experimental variant. Tab. 2/3: please report the number of replications of each variant. Tab. 3: Please explain the selection of these substrate mixtures. I cannot easily identify a systematic approach from this table. How did you develop the individual amounts of the single substrates, e.g., why did you chose 10, 30 and 60 % fly ash and not 5, 10, 15 or 40, 50, 60 % or something else? Did you carry out pre-experiments or are experiences available from preliminary tests? l. 162ff, results section general: In all figures standard deviation needs to be shown so that the degree of heterogeneity of the results becomes clearly visible. Results: it remains unclear why you group single experimental variants excluding others when comparing the plant performances. It would be interesting to know if there are other statistically significant differences between the test variants. I would like to suggest creating a matrix showing all variants and all detected differences at a glance. l. 249: as you distinguish between a results section and a discussion section you should avoid the interpretation of results here: “Worse” and “better” are terms of a valuation. In this section you should stick to neutral terms like “larger” or “smaller” etc. without further valuating these observations. l. 460: fly ash is usually characterized by a silty texture (see above). Thus, please provide a reference for your statement that ash has a coarse texture and is used for improving fine textured soils. Furthermore, please discuss the potential environmental hazards if fly ash is used as a topsoil substitute. E.g., in Germany the use of ashes as amendments for soils is generally prohibited due to frequently occurring high heavy metal contamination. l. 465: “Liebig” law, not “Leibig” l. 487: I mentioned above that the transfer of these results to the field scale seems to be problematic. Please discuss the limitations of transferability. l. 509: I cannot find any results on the number of plants in the experimental variants. Thus, these conclusions are somewhat vague. References: The sorting of the list of reference is not correct. Some authors are ordered by their first names instead of their names (e.g., Frank Nicolini => Nicolini, Frank)

Author Response

Dear Editors and reviewers:

On behalf of my co-authors, we thank you much for giving us an opportunity to revise our manuscript, we appreciate editor and reviewers very much for their positive and constructive comments and suggestions on our manuscript entitled “Responses of Melilotus.suffruticosa growth to the composition of the different topsoil substitute materials in the reclamation of open-pit mininggrassland area in Inner Mongolia” (ID: materials-604268).

We have made the point-by-point revisions according to the reviewers’ comments. We believe that the revised version may be of particular interest to the readers of your journal as it presents. The reviewers gave us very valuable comments and suggestions. We accepted the comments and suggestions and followed them. The reviewer comments are laid out below in italicized font and specific concerns have been numbered. Our response is given in normal font, at the same time, we have numbered each line of the revised manuscript. For details, please refer to the revised manuscript. The detailed point-by-point revision according to the reviewers’ comments are listed as follows:

Response to Reviewer 1 Comments

Point 1:First of all the language of the manuscripts needs a rigorous editing. Several terms need to be checked and seperated.

Response 1: We sincerely thank you for pointing this out. I'm sorry that the article have a spelling problem due to our carelessness, all the language of the manuscripts have done a rigorous editing.Several terms have been checked and seperated ,such as “iste”.

Point 2:,The described experiments need to be explained further in detail. E.g., the selection of the test plant species remains unclear: Is this species a native species of the described mining region in China and has it specific functions for reclamation? Can this species be regarded as exemplary for other species of the region?

Response 2: We sincerely thank you for pointing this out. First of all, this species is a legume herb with the effect of fertilizing the soil. Secondly, the characteristics of grass-tolerant, barren-resistant and alkali-tolerant soils are highly adaptable to the region. Finally, in the reclamation work before, the grass raft was also planted and grown well in the study area. Based on the above three points, this species is suitable as test species.

Point 3: l. 47: “well” mining: do you mean subsurface mining?

Response 3: It means underground mining.

Point 4: l. 75: as I know the conditions in the Rhenish Lignite Mining District very well, I found that the reference is wrongly cited: Loess deposits in Germany are of Quaternary age and must be considered as natural topsoil material, not as substitute of topsoil.

 Response 4: I read the document again and found that the reference is wrongly cited and deleted it..

Point 5: l. 95: “iste”? This term is used repeatedly throughout the text.

Response 5: This term is wrongly used,correct for “waste”

Point 6: l. 99: (see above), please explain here properly why this species is suitable as test species

Response 6: First of all, this species is a legume herb with the effect of fertilizing the soil. Secondly, the characteristics of grass-tolerant, barren-resistant and alkali-tolerant soils are highly adaptable to the region. Finally, in the reclamation work before, the grass raft was also planted and grown well in the study area. Based on the above three points, this species is suitable as test species.

Point 7: Tab. 1: please provide information about pH values of the substrates and of heavy metal contents. Particularly fly ash is usually problematic due to high pH values and higher heavy metal contents. In addition, please explain why fly ash had a sandy texture in your experiments. Usually fly ash is characterized by high amounts of silt (particles which are able to fly very well)

Response 7: We thank you very much for pointing this out. I am so sorry that due to the limitations of the soil testing equipment, it is impossible to test the pH values of the substitubes materials according to the method of testing the soil pH, so we may not be able to provide the pH of the substitubes materials.But But after the pot experiment, we tested the pH values of reconstructed soil in each pot.

NO

pH values

C1

6.30

C2

5.93

C3

6.69

C4

6.70

D1

7.37

H1

8.03

H2

7.87

H3

6.88

H4

6.97

H5

6.48

H6

7.29

H7

7.57

H8

7.33

H9

7.22

H10

4.08

H11

6.91

H12

7.03

H13

6.40

H14

7.35

H15

7.59

H16

7.64

H17

6.74

H18

7.40

We find that the pH values of some schemes do increase,but not much higher than D1.

As for the heavy metal content, these materials have been used in the reclamation work of the mine site during the investigation at the beginning of the project. In comparison, heavy metal pollution in coal mine is relatively small, so it was not considered at the beginning of the test. In the future work, we will make up for the shortcomings in this respect

Point 8: l. 127: please explain for which purposes these devices were used in your experiments or for laboratory analyses.

Response 8: CP114 electronic balance is used to measure the biomass,101-2AB electric heating blast dryer is used to dry grass samples,ruler is used to measure plant height, leaf width and leaf length,camera is used to photograph the growth

Point 9:l. 129 ff: Please discuss if the applied very shallow layers can be seen as representative or as comparable with field conditions. It can be assumed that roots are growing through such thin layers so that the layering does not make much sense. I would also expect that it will not be possible to create such thin layers under field conditions.

Response 9: We sincerely thanks for your professional comments. In the pre-sampling process, we found that the thickness of the topsoil is about 20cm in the undisturbed area, the following is the calcic horizon. The sampling of the reclaimed dumping site in the study area found that the thickness of the covering soil in most areas is about 10-15 cm. We have reason to believe that due to the randomness of construction, some areas will have a thickness of 5cm-10cm. Therefore, we want to use this experiment to see if the roots can penetrate into the vermiculite layer under such thin soil conditions, and whether the plants can continue to grow.

Point 10: l. 133: why do you call the layered variants “hierarchical”? I think this term does not fit here.

Response 10: We thank you very much for your comments for pointing out this omission. we have corrected the “Hierarchical” into “Layered variants schemes”

Point 11:l. 134: the “topsoil scheme” is not further explained. Please provide some basic information on the topsoil used for this experimental variant. Tab. 2/3: please report the number of replications of each variant. Tab. 3: Please explain the selection of these substrate mixtures. I cannot easily identify a systematic approach from this table. How did you develop the individual amounts of the single substrates, e.g., why did you chose 10, 30 and 60 % fly ash and not 5, 10, 15 or 40, 50, 60 % or something else? Did you carry out pre-experiments or are experiences available from preliminary tests?

Response 11: We thank you very much for pointing this out.Topsoil scheme refers to the scheme that only topsoil is used and no other substitute materials. The scheme is used as a control scheme. The Topsoil Used in the experiment is more than 20 cm soil collected in the field.

Each group was repeated three times.

Before conducting the experiments design, we consulted the relevant literature, combined with the test results of different materials, and had a preliminary understanding of different topsoil substitutes materials. Compared with topsoil, the rock and soil stripping is mainly due to poor nutrient status and large chunks of gravel; coal gangue has a large particle size, it will cause a large loss of water while the content of coal gangue is excessive, but coal gangue can improve the soil nutrient status.

Fly ash is poor in nutrient status and is often used as a modifier. However, in order to fully demonstrate that the fly ash will cause poor plant growth, 10%, 30% and 60% are selected. It was also found that under the condition that the content of fly ash was 60%, the leaves of the hibiscus were yellow and a large number of deaths occurred. This is also a prerequisite for our experimental design.
We grouped according to the proportion of different materials, and divided the proportions of certain materials into one group to explore the commonality of the influence of this material on plant growth and the difference of the influence of other materials on plant growth.

Point 12: l. 162ff, results section general: In all figures standard deviation needs to be shown so that the degree of heterogeneity of the results becomes clearly visible. Results: it remains unclear why you group single experimental variants excluding others when comparing the plant performances. It would be interesting to know if there are other statistically significant differences between the test variants. I would like to suggest creating a matrix showing all variants and all detected differences at a glance.

Response 12: We sincerely thanks for your professional comments, I agreed with you for this issue .Standard deviation lines are added in all figures.

We grouped according to the proportion of different materials, and divided the proportions of certain materials into one group to explore the commonality of the influence of this material on plant growth and the difference of the influence of other materials on plant growth.

Point 13: l. 249: as you distinguish between a results section and a discussion section you should avoid the interpretation of results here: “Worse” and “better” are terms of a valuation. In this section you should stick to neutral terms like “larger” or “smaller” etc. without further valuating these observations.

Response 13: We thank you very much for your comments for pointing out this omission. Now, in the revised manuscript, we have corrected it.

Point 14: 1. 460: fly ash is usually characterized by a silty texture (see above). Thus, please provide a reference for your statement that ash has a coarse texture and is used for improving fine textured soils. Furthermore, please discuss the potential environmental hazards if fly ash is used as a topsoil substitute. E.g., in Germany the use of ashes as amendments for soils is generally prohibited due to frequently occurring high heavy metal contamination.

Response 14: Fly ash has a coarse texture and is used for improving fine textured soils.

Wang Haizhen,Xu Jianming,Xie Zhengmiao,et al.Effects of fly ash on soil and the growth of crop[J].Soil and Environmental Sciences,1999,8(4):305-308.

Zhao Liang,Tang Zejun.Effect of soil physical properties using fly ash as sandy soil amendment[J].Journal of Soil and Water Conservation,2009,23(6):178-202.

Ghodrati M, Sims J T, Vasilas B L. Evaluation of fly ash as a soil amendment for the Atlantic coastal plain, Soil hydraulic properties and elemental leaching[J]. Water Soil Air Pollution, 1995, 81: 349-61.

The use of weathered fly ash can increase the accumulation of Se in crops. Some short-term indoor incubation experiments have found that adding unweathered fly ash to sandy soil can inhibit microbial respiration, enzyme activity and soil N cycling. Fly ash contains 5%~30% toxic elements, especially Cd, Cu and Pb can be filtered out, which may cause soil, water and biological pollution, especially the high content of soluble salt in weathered fly ash, which is more likely to cause groundwater pollution.

Stoewsand G S, Gutenmann W H, Lisk D J. Wheat grown on fly ash: high selenium uptake and response when fed to Japanese quail[J]. Journal of Agricultural and Food Chemistry, 1978, 26: 757-759.

Garau M A, Dalmau J L. Felipo M T. Nitrogen mineralization in soil amended with sewage sludge and fly ash[J]. Biology and Fertility of Soils, 1991, 12: 199-201.

Lai K M, Ye D Y, Wong J W C. Enzyme activities in a sandy soil amended with sewage sludge and coal fly ash[J]. Water Air Soil Pollution, 1999, 113: 261-272.

Cothern C R, Smith JE. Environmental Radon[M]. New York:Plenum Press, 1987: 363.

Point 15:,l. 465: “Liebig” law, not “Leibig”

Response 15: We were really sorry for our careless mistakes. Thank you for your reminding,we have corrected the “Leibig” into “Liebig”.

Point 16:,l. 487: I mentioned above that the transfer of these results to the field scale seems to be problematic. Please discuss the limitations of transferability.

Response 16: According to your suggestion , we discuss the limitations of transferability Pot experiment can only simulate the conditions of field crop growth to the maximum extent, it can not achieve complete consistency, so it will cause differences in crop growth. In pot experiment, the mixing of different materials is more uniform, but in field work, this effect can not be achieved. In field experiment, we found that there may be only one material in some areas, but no mixing. In the actual construction process, the compaction of reconstructed soil by large-scale machinery will result in the increase of bulk density and the decrease of porosity, which will inevitably affect the growth of plants.

Point 17: l. 509: I cannot find any results on the number of plants in the experimental variants. Thus, these conclusions are somewhat vague.

Response 17: We sincerely thanks for your suggestion. The number of plants in the potted plants is not indicated in the text. This result is derived from the data collected by the camera. The number of plants in the potted plants can be seen from the photographs, so this conclusion is reached. But this result is more due to our subjective knowledge, so it was revised. There are three cases, the biomass is large, the growth of individual plants is also better; the biomass is small and the growth of single plants is better; the biomass is large and the growth of individual plants is poor.

Point 18: References: The sorting of the list of reference is not correct. Some authors are ordered by their first names instead of their names (e.g., Frank Nicolini => Nicolini, Frank)

Response 18: Thank you for your comments, the references have been reordered.

We hope that these revisions are satisfactory and that the revised version will be acceptable for publication in Materials.

Thank you very much for your work concerning my paper.

Wish you all the best!

Reviewer 2 Report

Even if the paper addresses an interesting topic related to soil reuse and derelict land reclamation, there are several issues that were not adequately addressed. 

Literature review misses several seminal works and important advances crossing top soil adaptation for vegetation growth. Read for example (Burley et. al regarding this subject).

Material and methods presents several flaws, considering not only that it is hard to understand the used methodology, but also because the research steps are not adequately described.

This area needs to be corrected and the introduction of a phased methodological diagram is considered crucial even to increase researchers awareness of the methodological flaws.

The results are also not adequately presented. Further information is needed. Discussion needs to be supported on further data.

Conclusions need to be more scientific. As they are they highlight the limitations of the research.

Author Response

Dear Editors and reviewers:

On behalf of my co-authors, we thank you much for giving us an opportunity to revise our manuscript, we appreciate editor and reviewers very much for their positive and constructive comments and suggestions on our manuscript entitled “Responses of Melilotus.suffruticosa growth to the composition of the different topsoil substitute materials in the reclamation of open-pit mininggrassland area in Inner Mongolia” (ID: materials-604268).

We have made the point-by-point revisions according to the reviewers’ comments. We believe that the revised version may be of particular interest to the readers of your journal as it presents. The reviewers gave us very valuable comments and suggestions. We accepted the comments and suggestions and followed them. The reviewer comments are laid out below in italicized font and specific concerns have been numbered. Our response is given in normal font, at the same time, we have numbered each line of the revised manuscript. For details, please refer to the revised manuscript. The detailed point-by-point revision according to the reviewers’ comments are listed as follows:

Response to Reviewer 2 Comments

Point 1:Literature review misses several seminal works and important advances crossing top soil adaptation for vegetation growth. Read for example (Burley et. al regarding this subject).

 Response 1:

Based on this question,we add some references.

Burley J B, Thomsen C H, Kenkel N. Productivity equation for reclaiming surface mines[J]. Environmental Management, 1989, 13(5):631-638.

Burley J B, Thomsen C H. Application of an agricultural soil productivity equation for reclaiming surface mines: Clay County, Minnesota[J]. International Journal of Surface Mining Reclamation & Environment, 1990, 4(3):139-144.

Bryan Burley J , Fowler G W , Polakowski K , et al. Soil Based Vegetation Productivity Model for the North Dakota Coal Mining Region[J]. International journal of surface mining, reclamation and environment, 2001, 15(4):213-234.

Point 2:Material and methods presents several flaws, considering not only that it is hard to understand the used methodology, but also because the research steps are not adequately described.This area needs to be corrected and the introduction of a phased methodological diagram is considered crucial even to increase researchers awareness of the methodological flaws.

Response 2: We thank you very much for pointing this out. Before conducting the experiments design, we consulted the relevant literature, combined with the test results of different materials, and had a preliminary understanding of different topsoil substitutes materials. Compared with topsoil, the rock and soil stripping is mainly due to poor nutrient status and large chunks of gravel; coal gangue has a large particle size, it will cause a large loss of water while the content of coal gangue is excessive, but coal gangue can improve the soil nutrient status.

Fly ash is poor in nutrient status and is often used as a modifier. However, in order to fully demonstrate that the fly ash will cause poor plant growth, 10%, 30% and 60% are selected. It was also found that under the condition that the content of fly ash was 60%, the leaves of the hibiscus were yellow and a large number of deaths occurred. This is also a prerequisite for our experimental design.

We grouped according to the proportion of different materials, and divided the proportions of certain materials into one group to explore the commonality of the influence of this material on plant growth and the difference of the influence of other materials on plant growth.

Point 3:The results are also not adequately presented. Further information is needed.

Response 3:

We sincerely thanks for your professional comments, I agreed with you for this issue, so we add more information in the results section.In all figures standard deviation lines have been added,so that the degree of heterogeneity of the results becomes clearly visible.

Point 4:Discussion needs to be supported on further data.

Response 4: According to your suggestion,data were added to support the discussion, as follows:

Reasons for the differencesin the physicochemical properties of reconstructed soils and the growth status of Melilotus.suffruticosa

The theory of soil-forming factors posits that soil is the product of multiple natural factors, such as biology, climate, parent material, topography, and time, as well as human activities. In this study, based on the theory of soil-forming factors and the reconstruction of soil parent material, different soil profiles were reconstructed using solid wastes (coal gangue, rock and soil stripping materials, and fly ash) generated during mining as surface soil substitutes. Dong (2008) and other studies have shown that the physicochemical properties of soils that developed from different parent materials (bedrock) are quite different. For instance, coal gangue has a large particle size and high organic matter content(Fan et al.,2003;Jiang,Li,1998), fly ash has a large particle size(Gao et al.,2003), is hydrophilic(Yang et al.,1997), and has poor nutrient status(Yang et al.,2000), and the physical properties of rock and soil exfoliates from stripping are similar to those of the original topsoil, but with a lower organic matter content,as the organic matter content, total nitrogen content, available phosphorus and available potassium content of the rock and soil stripping are more than double that of the topsoil.Therefore, the physicochemical properties of reconstructed soils vary greatly if different combinations of materials are used.

The effects of including materials in different ratios on the production of plant biomass were greater than those of these differences on leaf width, plant height, and root length. During the experiment, it was found that the mixture schemes resulting in lower biomass tended to have fewer plants and lower coverage, but the growth of a single plant in these cases was sometimes better; further, the schemes resulting in lower biomass but more plants tended to have poor performance in terms of leaf width or plant height.The reason for this result is that there were differences in the proportions of different components included in the soil substitute in these different schemes, resulting in differences in their physicochemical properties. Different endowments of soil resources result in different productivity of the replacement soil. Therefore, in the actual process of vegetation reconstruction, the use of a reasonable planting density and topdressing will also play important roles in the successful growth of Melilotus.suffruticosa.

The number of plants in the potted plants is not indicated in the text. This result is derived from the data collected by the camera. The number of plants can be seen from the photographs, so this conclusion is reached. But this result is more due to our subjective knowledge.

Reasons for the better growth of Melilotus.suffruticosa under specific substitute material ratios

When the coal gangue content of the mixture used was below 30% or 10%, the growth of Melilotus.suffruticosa performed better in terms of all four indicators assessed. According to the physical and chemical properties of solid waste, the average organic matter content of coal gangue is 43.9g/kg, which is about 25% higher than the organic matter content of topsoil, and the available K content of coal gangue is about 10% higher than that of topsoil. Coal gangue has a large particle size and large pores,its particle size is 2-5cm。When stacked, there are many and large pores between coal gangue particles. This characteristic means that, when coal gangue is piled up, moisture can easily seep downward through it(Guo et al.,2008), so the content of coal gangue in the soil replacement material should not be too high. However, including coal gangue can improve soil nutrient status(Duan,et al.,1999). At the same time, as the degree of weathering increases, the sizes of the internal weathering cracks in coal gangue increase, which makes coal gangue have more water-holding capacity(Cai et al.,2015) and thus be more conducive to the growth of vegetation. However, in the actual production process, coal gangue is generally excessively produced, where as the availability of the surface soil is limited, so the optimal amount of coal gangue used should be about 30%. When layered and overlapped coal gangue and reconstructed soil or topsoil is used, the roots of Melilotus.suffruticosa can penetrate into the coal gangue layer,the root length of Melilotus.suffruticosa is about 11-12 cm.. The development of crevices in the coal gangue layer can provide the necessary water and nutrients for M.suffruticosa, helping this plant to grow better.

Fly ash is mainly used to improve the physical structure of clay because of its large particle size, but clay did not exist in the sandy loam soil used in this study’s experiments. The hydrophilicity of fly ash has little effect on plant growth under conditionsin which an adequate supply of the water required for the growth of vegetation is available or provided. Fly ash can reduce soil bulk density, increase porosity, adjust three-phase ratio and increase ground temperature. If used to improve sandy soil, it can increase water holding capacity and hydraulic conductivity, which is helpful to prevent crust.However, due to the fact that there is almost no nitrogen in fly ash, it was also found that the leaves of Melilotus.suffruticosa began yellowing during the experiment when this material was included. According to Leibig’slaw of the minimum, this characteristic thus became a limiting factor for the growth of Melilotus.suffruticosa. That is to say, under the conditions of an indoor pot experiment, as the fly ash content of the mixture increases, the soil nutrient condition becomes worse, resulting in lessproduction of Melilotus.suffruticosa biomass. In 2008, Chen Jing et al.(2008) also showed that with anincrease in fly ash consumption, the growth of plants deteriorated.

The use of weathered fly ash can increase the accumulation of Se in crops. Some short-term indoor incubation experiments have found that adding unweathered fly ash to sandy soil can inhibit microbial respiration, enzyme activity and soil N cycling. Fly ash contains 5%~30% toxic elements, especially Cd, Cu and Pb can be filtered out, which may cause soil, water and biological pollution, especially the high content of soluble salt in weathered fly ash, which is more likely to cause groundwater pollution.

However, according to the pot experiment results of this study, when the content of fly ash was below 10%, it had little effect on the growth of Melilotus.suffruticosa.

Benefits and limitations of using solid waste from the mining industry as a soil reconstruction material

The selection of a reasonable compositional scheme based on experimentsusing the different solid waste produced in mining processes as soil substitute materialsis of great significance for land reclamation in mining areas with scarce topsoil. Firstly, this allows the problem of poor vegetation growth caused by the insufficient thickness of the overlying soil in such areas to be solved. Secondly, the cost to mining enterprises of purchasing topsoil is greatly reduced, as reconstructing the soil by using substitute materials can save more than 50% of the available topsoil;for example, assuming asoil cover thickness of 30 cm, the total depth of the surface soil needed is thus less than 1500m per hectare, and if the local price of surface soil is about 30 yuan/m then in this case the reclamation investment cost is reduced by 45 000 yuan per hectare. Finally, the problem of the disposal of solid waste generated in the mining process is solved. It can thus be seen that using solid waste as a substitute material for topsoil has certain economic and ecological benefits. However, at the same time, the current study’s results are based on laboratory tests, so their applicability in the field is not yet known. The next step is to carry out plot experiments in the field to determine the best application scheme.

Point 5:Conclusions need to be more scientific. As they are they highlight the limitations of the research.

 Response 5:

Once again, we acknowledge your comments very much, which are valuable in improving the quality of our manuscript. Now, in the revised manuscript, we have corrected.

(1) Based on the present study’s results, the use of mining solid wastes as a substitute for topsoil to sustain plant growth appears to be feasible according to analyses of the four selected indices of the growth status of Melilotus.suffruticosa tested. This conclusion can help the owner of the mining area solve the problem of resource utilization of solid waste and scarcity of topsoil

(2) When the thickness of the reconstructed upper soil was greater than 10cm, the biomass of Melilotus.suffruticosa was higher than that obtained with pure topsoil, and when the reconstructed upper soil was placed above the natural soil, the biomass of Melilotus.suffruticosa obtained was higher than that obtained on topsoil alone. When the amount of coal gangue added was controlled to be 30% of the mixture, the biomass of Melilotus.suffruticosa obtained was the best, and the values of other growth indicators were also better. The overall biomass obtained with the mixture schemes containing fly ash was lower, and was obviously different from that obtained in the control scheme, but the growth of individual plants was better; therefore, the growth of Melilotus.suffruticosa was better when the amount of fly ash added was controlled to be 10% or less. When the content of rock and soil stripping materials from miningwas controlled to be 40% or less, Melilotus.suffruticosa showed good growth in terms of all four indicators. This scheme was thus more suitable for sustaining the growth of Melilotus.suffruticosa than all of the others tested. And when the ratio of topsoil:coalgangue:rock and soil stripping materials was 3:3:4, respectively, the biomass of Melilotus.suffruticosa was the highest, and the increases in leaf width, plant height, and root length were also better. But this conclusion comes from laboratory tests, and it is unknown whether it is applicable to the local area. This result only shows that these materials can be used as substitutes materials for topsoil. In different mining areas, the specific proportion needs to be tested before it can be obtained.

(3) The biomass of Melilotus.suffruticosasignificantly differed among the schemes tested, but the other three indicators did not significantly differ among schemes in many cases. In treatments in which plants had small biomass and also tended to produce fewer plants, the growth statuses of individual plants were better. Treatments that produced average amounts of plant biomass but more plants tended to have poor performance in terms of leaf width or plant height. Therefore, in the actual reclamation process, planting density should be reasonably arranged according to the physical and chemical properties of the soil. Of course, the best way is to reclaim land strictly according to the best proportion obtained from the experiment, but the economic and ecological conditions of different mining areas are different, so it is likely that the best proportion can not be chosen. Therefore, the combination of substitubes materials proportion and planting method can achieve the goal of reclamation better.

We hope that these revisions are satisfactory and that the revised version will be acceptable for publication in Materials.

Thank you very much for your work concerning my paper.

Wish you all the best!

Round 2

Reviewer 2 Report

The research continues to present the previously identified mistakes and flaws.

Though strengthening the literature review, the methodological flaws identified on the previous version are still present.

It is hard to understand the used methodological approach. 
